# Wildlife Dermatophytoses in Central Italy (Umbria and Marche Regions): A Fifteen-Year Investigation (2010–2024)

**DOI:** 10.3390/jof11100753

**Published:** 2025-10-21

**Authors:** Silvia Crotti, Deborah Cruciani, Nicoletta D’Avino, Alessandro Fiorucci, Giulia Morganti, Daniele Paoloni, Manuela Papini, Vincenzo Piscioneri, Alice Ranucci, Sara Spina, Marco Gobbi

**Affiliations:** 1Istituto Zooprofilattico Sperimentale dell’Umbria e delle Marche “Togo Rosati”, Via G. Salvemini 1, 06126 Perugia, Italy; s.crotti@izsum.it (S.C.); n.davino@izsum.it (N.D.); a.fiorucci@izsum.it (A.F.); v.piscioneri@izsum.it (V.P.); a.ranucci@izsum.it (A.R.); s.spina@izsum.it (S.S.); m.gobbi@izsum.it (M.G.); 2Department of Veterinary Medicine, University of Perugia, Via San Costanzo 4, 06126 Perugia, Italy; giulia.morganti@unipg.it; 3Istituto Oikos srl, Via Crescenzago 1, 20134 Milano, Italy; daniele.paoloni81@gmail.com; 4Clinica Dermatologica di Terni, Dipartimento di Medicina e Chirurgia, Università degli Studi di Perugia, 06123 Perugia, Italy; manuelapapini@tiscali.it

**Keywords:** *Arthroderma* spp., dermatophytes, *Microsporum* spp., *Paraphyton* spp., *Trichophyton* spp., wildlife rescue centers, veterinary hospitals

## Abstract

The expansion of urbanized areas is leading to increased human−animal interactions, thereby creating potential new ecosystems for wildlife. In this context, dermatophytoses are of particular significance. This investigation aimed to evaluate the presence of dermatophyte species in wild animals enrolled by passive control and targeted active control plans or deceased for causes independent of this study and analyzed for necroscopic procedures. From 2010 to 2024, a total of 704 samples were collected and analyzed by conventional and molecular assays. Dermatophytes were detected in 77 animals. The molecular approach identified *Paraphyton mirabile* (5.96%), *Trichophyton mentagrophytes* complex (2.41%), *Microsporum canis* (0.71%), *Arthroderma curreyi*, *A. thuringense* (0.57% each), *A. uncinatum*, and *A. eboreum* (0.43% each). In one animal a co-infection of *T. mentagrophytes* and *M. canis* was found. Considering wild animals as sentinels for dermatophytoses, it is crucial to implement surveillance to prevent potential zoonotic outbreaks.

## 1. Introduction

Urban ecosystems are transforming cities into novel habitats hosting diverse wildlife [1]. Wild animals increasingly serve as reservoirs of emerging zoonotic pathogens, posing risks to both human and animal health [2]. They can also act as bioindicators of endemic diseases (e.g., tuberculosis and brucellosis) that might otherwise remain subclinical in livestock [3]. Moreover, their health reflects ecosystem health, making them crucial for environmental monitoring [4]. For these reasons, Italy, along with many other European countries, has established national plans to assess the health status of wildlife, focusing on the protection of public health, livestock, domestic animals, and ecosystems of which wild fauna are an integral part in a One-Health perspective [5]. Moreover, wildlife rescue centers (WRCs) and veterinary hospitals (VHs) in local settings play a crucial role in the rescue and rehabilitation of wild animals. These institutions actively collaborate with diagnostic labs to detect infectious diseases, particularly those with zoonotic potential, and monitoring the antimicrobial resistance [6,7].

Dermatophytes are among the most prevalent causes of human and animal mycoses: particularly, changes in the pattern of human interaction with animals, including pets, farm, and wild animals, have an impact on changes in the epidemiology of dermatophytoses [8]. According to the literature, fungi belonging to the *Trichophyton mentagrophytes* complex, *Microsporum canis*, and *Nannizzia gypsea* are the most common found in a wide range of wild animals. In addition, *N. persicolor* is primarily carried by sylvatic rodents and *T. erinacei* is the most common causative agent of ringworm in wild hedgehogs [8]. Dermatophytes may be transmitted from wild animals to humans via direct contact or indirectly from the environment, as infectious propagules (e.g., arthroconidia) can remain viable for years under optimal temperature and humidity conditions [9]. Therefore, attention should be paid to professionals who closely interact with wildlife (e.g., wildlife rehabilitators, veterinarians, and technicians), and especially to ordinary citizens who might encounter wild animals inhabiting gardens or public parks [10]. Moreover, it should be noted that hunting pets may act as carriers after previous contact with wild animals and their habitats and can subsequently infect humans in both urban and rural settings. In some cases, wild animals are raised as pets, like hedgehogs in Japan [11] and tourism could be another source of human exposure to dermatophytes [12]. Since the zoonotic potential of several dermatophytes species is not equal, the prevalence of human infections varies. Considering that animal dermatophytoses are mainly caused by zoophilic and geophilic dermatophytes, the zoophilic species *M. canis* and *T. mentagrophytes*, and the geophilic *N. gypsea* are the most frequently transmitted to humans. According to Švarcová et al., *T. mentagrophytes* complex includes a variety rank which can be distinguished only by unambiguous molecular identification [13]. Moreover, among *T. mentagrophytes* var. *mentagrophytes* and *T. mentagrophytes* var. *interdigitale*, different ITS genotypes can be distinguished [14].

Even if considered mild infections, dermatophytes may induce severe inflammations resulting in itching, burning, sleep disturbance, and even depressive states, which substantially impair patients’ quality of life [15]. Such effects tend to be particularly pronounced in humans, who, lacking a dense fur coat, may suffer more severe reactions to zoophilic or geophilic dermatophyte infections than animals [16].

In the literature, few data on wildlife dermatophytoses are available [17,18]. The aim of this study was to investigate the presence of dermatophytes in wild fauna and perform molecular characterization of the isolates.

## 2. Materials and Methods

Between 2010 and 2024, a total of 704 wild animals from central Italy, specifically Umbria and Marche regions, were enrolled in this investigation, and their annual distribution is shown in Figure 1.

In detail, a dual approach was adopted combining passive surveillance with targeted active surveillance plans. Passive surveillance involved locals institutions such as WRCs, the University of Perugia Veterinary Teaching Hospital, and the Istituto Zooprofilattico Sperimentale dell’Umbria e delle Marche “Togo Rosati” (IZSUM) for monitoring sick or deceased wild animals to identify health trends and detect unknown or emerging diseases [19]. Meanwhile, the targeted active surveillance focused on specific wildlife species; for instance, the European LIFE “U-SAVEREDS” project aimed to understand and mitigate infectious disease risks in grey squirrel populations [20].

### 2.1. Animals

#### 2.1.1. Squirrels

In the context of the LIFE “U-SAVEREDS” project (2015–2018), preceded by the RESCO regional control plan (2012–2013), 257 suppressed grey squirrels (*Sciurus carolinensis* Gmelin, 1788) and 1 captured and released red squirrel (*Sciurus vulgaris* Linnaeus, 1758) were enrolled in the present study. These animals were sampled in the management units (MUs) established among the LIFE “U-SAVEREDS” project, as previously reported [20]. In addition, 6 red squirrels sampled among passive surveillance were included in the study (Figure 2). One of the latter 6 was alive at the time of sampling, while the remaining 5 had died from causes independent of this study and were sampled during necropsy procedures performed at the IZSUM.

#### 2.1.2. European Hedgehogs

One hundred sixteen wild European hedgehogs (*Erinaceus europaeus* Linnaeus, 1758) rescued by WRCs or admitted to VHs for emergency care were enrolled in the survey (Figure 2). Forty-eight animals were alive at the time of sampling, while the remaining sixty-eight were deceased as a result of causes independent of this study and were sampled during necroscopic procedures (IZSUM).

#### 2.1.3. Other Wild Fauna

The other animals enrolled in this investigation were deceased and directly conferred to the IZSUM in order to perform necroscopic procedures. In detail, they were: 108 roe deer (*Capreolus capreolus* Linnaeus, 1758), 96 red foxes (*Vulpes vulpes* Linnaeus, 1758), 40 grey wolves (*Canis lupus* Linnaeus, 1758), 33 European badgers (*Meles meles* Linnaeus, 1758), 30 crested porcupines (*Hystrix cristata* Linnaeus, 1758), 6 pine martens (*Martes martes* Linnaeus, 1758), 4 beech martens (*Martes foina* Erxleben, 1777), 3 European wild cats (*Felis silvestris* Schreber, 1777), 2 fallow deer (*Dama dama* Linnaeus, 1758), 1 red deer (*Cervus elaphus* Linnaeus, 1758), and 1 Eurasian beaver (*Castor fiber* Linnaeus, 1758) (Figure 2).

### 2.2. Bioethics

Grey squirrel samples enrolled during the European LIFE “U-SAVEREDS” project and included in this study were collected following the management plan in accordance with the 92/43/CEE Directive, European Union Regulation No. 1143/2014, and the National Hunting Law 157/92 (Italy), with positive evaluation by ISPRA (ID: 14163/A4C/2016).

No prior approval from an ethics committee was required to collect samples from live wild animals as the procedures used did not cause pain, suffering, distress, or lasting harm beyond that of a standard needle injection, and were therefore in line with accepted veterinary best practices [21].

### 2.3. Sampling

Samples for dermatophyte investigations were represented by fur and quills carried out through the Mackenzie technique using a toothbrush or a scalp brush, based on the size of the animal. Samples from live animals were collected by gently brushing to avoid stressful procedures; samples from animals deceased were taken by brushing all the available surfaces. Most sampled animals appeared to be healthy and free from macroscopic external lesions, except 5 of them (0.71%). In detail, 4 grey squirrels of the LIFE “U-SAVEREDS” project showed non-exudative alopecic lesions on the back (Figure 3a), paw (Figure 3b), muzzle, and tail, respectively. The red squirrel admitted at the University of Perugia Veterinary Teaching Hospital showed widespread crusted dermatitis and alopecic skin lesions (Figure 3c).

Samples were collected in sterile tubes and delivered to the IZSUM mycology laboratory, without any substance used as the transport medium.

### 2.4. Cultural and Molecular Identification

#### 2.4.1. Wild Ruminants

Samples of roe deer, fallow deer, and red deer were directly subjected to DNA extraction using the QIAamp DNA mini kit (QIAGEN^®^, Hilden, Germany) following a modified Gram-positive protocol (Appendix D: Protocols for Bacteria, Isolation of genomic DNA from Gram-positive bacteria), given the unavailability of Dermasel Agar supplemented with thiamine and inositol, required from typical dermatophytes isolated in wild ruminants, such as *Trichophyton verrucosum*. The extracted DNA was analyzed using a hemi-nested PCR assay. In the first step, primers DMTF18SF1 and DMTF28SR1 amplified conserved regions in the 18S and 28S rRNA genes to detect a wide range of fungal species [22]; in the second step, primers DMTF18SF1 and DMTFITS1R were employed to specifically amplify the 5.8S rRNA gene of dermatophytes [23]. To guarantee the reliability of the PCR assay and avoid risk of contamination, in every analytical session, both positive and negative controls were included during the extraction and amplification phases.

#### 2.4.2. Other Wild Fauna

Samples collected from non-ruminant wild animals were inoculated onto Dermasel agar, incubated at 25 ± 1 °C, and observed daily for 14 days. Dermatophyte detection was based on macroscopic and microscopic features, supported by the mycology identification keys provided by the University of Adelaide [24]. Dermatophyte colonies were subjected to DNA extraction by the same protocol used for wild ruminants. An end-point PCR assay using universal fungal primers ITS1 and ITS4 was performed to amplify the ITS1–5.8S–ITS2 region flanked by conserved regions of the 18S and 28S rRNA genes [25].

PCR-positive products were purified using the QIAquick PCR Purification Kit (QIAGEN^®^). Sequencing was performed using the BrilliantDye^TM^ Terminator v3.1 Cycle Sequencing Kit (NimaGen^®^, Nijmegen, The Netherlands), and reactions were separated through 3500 Genetic Analyzer (Applied Biosystems^®^, Waltham, MA, USA). Consensus sequences were created by BioEdit Sequence Alignment Editor software, version 7.2.5 [26], and aligned in the Westerdijk Fungal Biodiversity Institute database [27]. Direct alignment proved reliable to identify dermatophyte species, except for differentiating species within the *T. mentagrophytes* complex. Therefore, to distinguish *T. mentagrophytes* var. *mentagrophytes* from *T. mentagrophytes* var. *interdigitale*, a single nucleotide polymorphism at position 94 was evaluated [28]. Moreover, based on the works by Nenoff et al. [29] and Taghipour et al. [30], each *T. mentagrophytes* var. *mentagrophytes* and *T. mentagrophytes* var. *interdigitale* strain was associated with a specific genotype.

## 3. Results

All 111 wild ruminants analyzed during the 15-year investigation period were negative at hemi-nested PCR. Through the ITS1/ITS4 PCR assay, dermatophytes were found in 77 (10.94%) other wild fauna. Figure 4 shows the annual distribution of positive and negative samples.

Positive samples belonged to the following non-ruminant wild fauna: grey squirrel (*n* = 60, 8.52%), red fox (*n* = 8, 1.14%), red squirrel (*n* = 6, 0.85%), European badger (*n* = 1, 0.14%), European hedgehog (*n* = 1, 0.14%), and crested porcupine (*n* = 1, 0.14%). In detail, conventional methods supported by molecular-based techniques allowed for the identification of seven different species: *Paraphyton mirabile* (*n* = 42, 5.96%), *Trichophyton mentagrophytes* complex (*n* = 17, 2.41%), *Microsporum canis* (*n* = 5, 0.71%), *Arthroderma thuringiense* (*n* = 4, 0.57%), *Arthroderma curreyi* (*n* = 4, 0.57%), *Arthroderma uncinatum* (*n* = 3, 0.43%), and *Arthroderma eboreum* (*n* = 3, 0.43%). A *T. mentagrophytes* complex isolate and a *M. canis* isolate were both recovered from the same individual—a red squirrel (Figure 5, Appendix A).

Recently, additional molecular investigations were performed on the strains belonging to the *T. mentagrophytes* complex. Unfortunately, the DNA was no longer available for 6 *T. mentagrophytes* strains isolated from 2 red foxes and 4 grey squirrels. Therefore, the additional analyses could be performed in the remaining 11 *T. mentagrophytes* strains belonging to 6 red squirrels and 5 grey squirrels. One red squirrel and one grey squirrel were infected with *T. mentagrophytes* var. *mentagrophytes* genotype III*, while the other 5 red squirrels and 4 grey squirrels harbored *T. mentagrophytes* var. *interdigitale* genotype II* (Appendix A).

Regarding symptomatic animals, the red squirrel was infected by *T. mentagrophytes* var. *mentagrophytes* genotype III*. It was admitted to the University of Perugia Veterinary Teaching Hospital and due to the severity and extent of its lesions, it received oral antifungal therapy. After two consecutive negative fungal cultures, taken at 1–2-week intervals, treatment was discontinued. The squirrel’s clinical condition improved, and the lesions fully resolved within approximately two months. The other symptomatic animals comprised 4 grey squirrels suppressed during the LIFE “U-SAVEREDS” project. Among these, *T. mentagrophytes* var. *interdigitale* genotype II* was identified in two individuals, while *P. mirabile* and *A. thuringense* were found in one animal each.

## 4. Discussion

This study (January 2010–December 2024) examined dermatophytes in 704 wild animals from Central Italy, using both passive surveillance (e.g., WildSENTINEL and necropsies) and active programs (e.g., RESCO and LIFE U-SAVEREDS). Therefore, sampling was not uniform over time, but active surveillance enabled extensive analysis of grey squirrels. A key goal of these programs was to remove the invasive Eastern grey squirrel, which competes with the native Eurasian red squirrel [31].

Except for wild ruminants that were directly analyzed using molecular techniques, a conventional approach combined with molecular analyses was employed in this study. Dermatophytes were found in 77 (10.94%) non-ruminant wild animals. Morphological features are still essential to isolate one or more than one fungal species, allowing for the detection of co-infection cases. In fact, in this study, a red squirrel was simultaneously infected by *T. mentagrophytes* complex and *M. canis*. However, cultural examination appears limited and molecular tools should be used to obtain a reliable identification of the dermatophyte species, whose taxonomic classification is in constant evolution. PCR assay and Sanger sequencing also allowed for identifying uncommon but not-negligible species. Thanks to this strategy, 7 species belonging to 4 different genera including *Paraphyton*, *Trichophyton*, *Arthroderma*, and *Microsporum* were detected [32]. The results obtained substantially agree with the literature on wildlife, except for the *Nannizzia* genus, which was not found in this investigation but was frequently reported in European countries, particularly in Italy [8].

*Paraphyton mirabile* was detected in 41 grey squirrels and in 1 European hedgehog representing the main dermatophyte isolated in this study, amounting to about 60%. *Paraphyton mirabile* has been already isolated in Italy, from horse and asymptomatic chamois, proving its presence in the wild [33,34]. The presence of *P. mirabile* in one hedgehog (0.14%) was surprising as dermatophytes exhibit a species-specific host range and the most common causative agent of ringworm in wild hedgehogs is *T. erinacei* [18]. In Poland, Gnat et al. reported a higher percentage of *P. mirabile* in European hedgehogs (6.0%) [10]. The identification of *P. mirabile* in wildlife is interesting because its ecological classification is still not clear [32,35]. Furthermore, it was isolated in a human patient affected by onychomycosis and *tinea pedis*, highlighting its potential zoonotic role [35].

Seventeen fungal strains belonging to the *T. mentagrophytes* complex were isolated from grey squirrels (*n* = 9), red squirrels (*n* = 6), and red foxes (*n* = 2). Additional molecular investigations conducted on 11 of these strains allowed for identifying *T. mentagrophytes* var. *interdigitale* genotype II* and *T. mentagrophytes* var. *mentagrophytes* genotype III*. The anthropophilic *T. mentagrophytes* var. *interdigitale* is suspected to have evolved from zoophilic *T. mentagrophytes* var. *mentagrophytes* often found on rabbits [36]. Moreover, genotype II* exhibites overlapping features of both *varietas* and is considered a mixed type primarily causing *tinea capitis* and *tinea faciei* in humans; it is spread in Europe, Asia, and Australia. Genotype III* is among the most prevalent in Europe, but it is reported in Canada, Japan, and India. It is primarily associated with zoonotic transmission, particularly from cats and rodents, and it has been reported to induce moderate to severe inflammatory lesions in humans, whereas those caused by other genotypes are typically milder [30]. This may pose a public health risk through environmental contamination or direct contact, particularly in grey squirrels given their confident behavior with humans [37].

*Microsporum canis* was detected in 4 red foxes and in the red squirrel co-infection case. This species has notably been described in domestic and stray cats, but infections linked to contact with symptomatic dogs have also been reported [38]. Rodents like guinea pigs are reported as animal hosts for this species less frequently [39]. Therefore, *M. canis* detection in other canids (e.g., red fox) and rodents was not surprising. This dermatophyte has demonstrated potential for both zoonotic and human-to-human transmissions [16]. In humans, clinical manifestation could be severe, particularly in children.

The genus *Arthroderma* encompasses geophilic dermatophyte species including *A. thuringense*, *A. curreyi*, *A. uncinatum*, and *A. eboreum* that were identified in a wide variety of wildlife through this investigation. In detail, *A. thuringense*, *A. curreyi*, and *A. uncinatum* had a higher frequency in a lower variety of animal species—grey squirrel and red fox—while *A. eboreum* had a lower frequency in a higher variety of animals—red fox, European badger, and crested porcupine. However, as reported in the literature, *A. eboreum* is a geophylic fungus already isolated from the soil of badger and rabbit burrows [40] and from the skin of human patients, highlighting its potential zoonotic role [41,42].

Squirrel samples showed the highest percentage of positivity (*n* = 66, 9.38%): overcrowding conditions may have caused grey squirrels to be more susceptible to infectious diseases, like dermatophytosis; however, it must be considered that the majority of samples were collected particularly from them (*n* = 264, 37.5%). In a similar study conducted in Iran, 3.14% of squirrels tested positive for *M. canis*, *Nannizzia gypsea* (formerly *M. gypseum*), *Lophophyton gallinae* (formerly *M. gallinae*), and *M. persicolor* [43]; *P. cookei* was instead isolated from grey squirrels in a similar study conducted in Italy [44]. However, it is not easy to compare data obtained on squirrels because most studies on rodents mainly focus on rabbits, hamsters, and guinea pigs.

Out of the 704 animals sampled, only 5 of them (0.71%), represented by 4 grey squirrels and 1 red squirrel, showed clinical lesions. The involvement of the *T. mentagrophytes* complex in three out of five lesion cases underscores its significant pathogenic potential. The absence of clinical lesions in the majority of wildlife indicates their potential role as asymptomatic carriers. Considering that positive samples constituted approximately 10% of the total analyzed samples, and that the animals were handled by sanitary personnel, the absence of human mycoses emphasizes the critical importance of wearing personal protective equipment (e.g., disposable gloves).

Despite the substantial number of animals analyzed, this study has several limitations. Sampling was geographically restricted to two central Italian regions, with the LIFE U-SAVEREDS project focused solely in Umbria—leading to a large number of grey squirrels and potential sampling bias. This concentration, while informative for that species, does not aim for statistical comparisons with less-represented animals.

## 5. Conclusions

Over 15 years, this study enhance the current understanding of the sentinel role of wild fauna in infectious disease surveillance—especially dermatophytoses—under a One Health framework integrating human, animal, and environmental health. Despite significant constraints such as biased data, irregular and variable-quality sampling, diagnostics adapted from domestic species, and limited stakeholder cooperation, wildlife surveillance remains relevant for the early detection of emerging or re-emerging pathogens. Conservation initiatives (e.g., the LIFE U-SAVEREDS project) and Wildlife Rescue Centers play a pivotal role in health monitoring and zoonotic risk assessment. The results obtained through this investigation revealed several dermatophyte species—including non-conventional ones—in wildlife from Umbria and Marche, notably *T. mentagrophytes* var. *mentagrophytes* genotype III* and *M. canis*, which are pathogenic to humans. Therefore, enhanced surveillance is critical to prevent zoonotic outbreaks.

## Figures and Tables

**Figure 1 jof-11-00753-f001:**
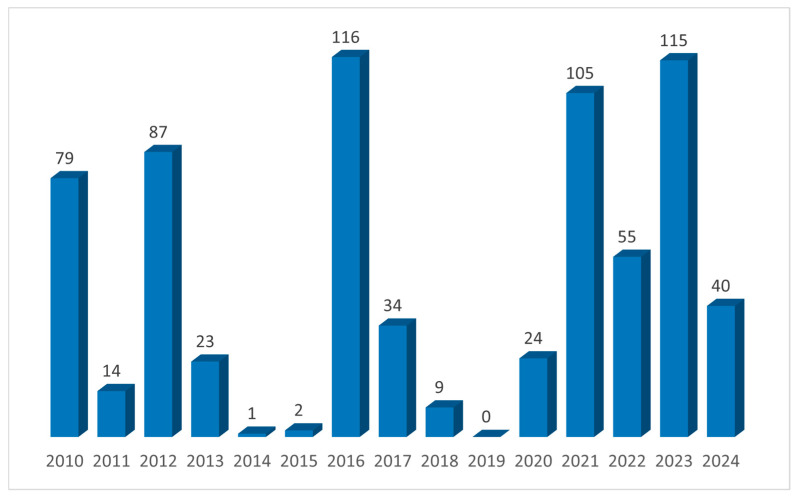
Annual distribution of wild animals sampled from 2010 to 2024.

**Figure 2 jof-11-00753-f002:**
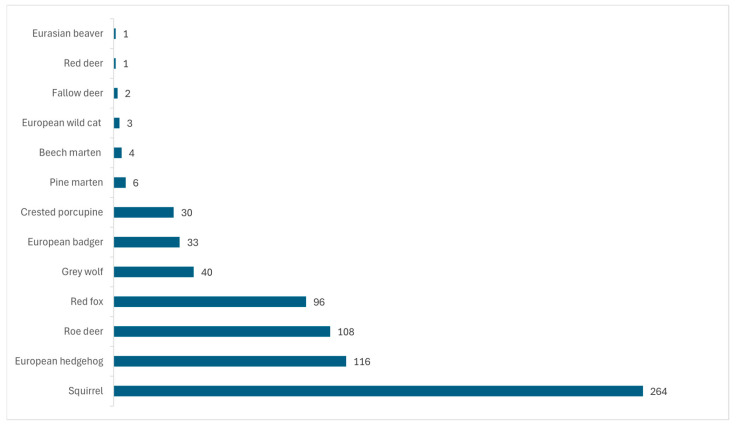
Wild fauna enrolled in this study between 2010 and 2024: the numbers on the right of the bars represent the amount of each species sampled.

**Figure 3 jof-11-00753-f003:**
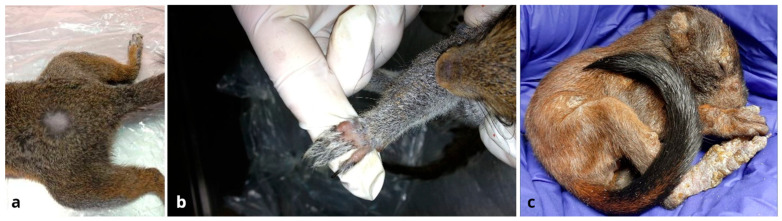
Non-exudative alopecic lesions on the back and paw of two grey squirrels (**a**,**b**) and widespread dermatitis in a red squirrel (**c**).

**Figure 4 jof-11-00753-f004:**
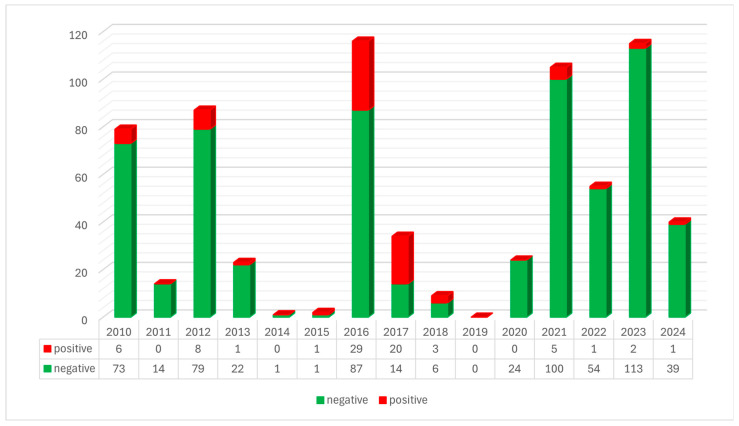
Annual distribution of positive and negative samples between 2010 and 2024.

**Figure 5 jof-11-00753-f005:**
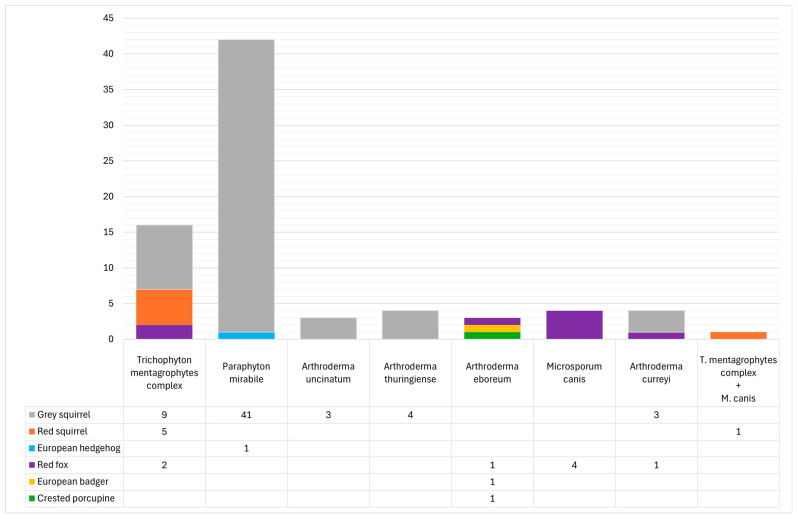
Details of dermatophytes species found in different animal species.

## Data Availability

The original contributions presented in this study are included in the article/Appendix A. Further inquiries can be directed to the corresponding author.

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
