# Peer review of "Wildlife Dermatophytoses in Central Italy (Umbria and Marche Regions): A Fifteen-Year Investigation (2010–2024)"

_jof, 2025, doi:10.3390/jof11100753_

Round 1

Reviewer 1 Report (New Reviewer)

The manuscript contains several data discrepancies that must be resolved. The percentages reported in the Results and Discussion sections use different denominators, leading to confusion. Additionally, the total number of isolates listed (n=76) does not match the number of positive animals (n=77), and there's a discrepancy in the total count of T. mentagrophytes strains between different parts of the manuscript (16 vs. 17). A taxonomic error was also found: the common names for the Martes species are incorrectly listed. Martes foina is the beech marten, and Martes martes is the pine marten. These must be corrected to ensure scientific accuracy. Finally, the manuscript states that the study was conducted until 2024. This is a future date and needs to be corrected to the actual end date of the study period.

The manuscript is comprehensive but could be more concise. Many sentences are long and complex, making the text difficult to follow. I suggest breaking them down into shorter, more direct sentences. Some word choices and phrasing also need revision for a more formal, scientific tone, such as replacing "vital players" with "crucial actors." There are also typos to be corrected, including "out-breaks" and "ivestock."

In the Materials and Methods section, there's a major point to be corrected: the inversion of the common names for the Martes species, where Martes foina is the beech marten and Martes martes is the pine marten. The linguistic precision also needs attention, with a revision of phrases like "were collected basing on" and a substitution of the verb "conferred." The term "grattacapo" should be translated or explained.

In the Results, there are significant numerical inconsistencies. The percentages for species are based on the total number of animals (704), while the overall positivity rate is based on non-ruminant animals (593). It is crucial to use a consistent denominator. The number of isolates (76) does not correspond to the number of positive animals (77), and the count of T. mentagrophytes strains (16) conflicts with the number used for further analysis (17). Please clearly differentiate between "positive animals" and the "number of isolates."

In the Discussion and Conclusion, the date inconsistency (study conducted until 2024) must be corrected. The percentages in the discussion do not match those in the results. The language, when presenting hypotheses, should be more cautious (for example, "overcrowding may have..." instead of "probably overcrowding"). The conclusion is long and repeats points already covered; please make it more concise and focused on the key takeaways.

Author Response

Point-by-point response to Comments and Suggestions for Authors

Major comments

Comments 1: The manuscript contains several data discrepancies that must be resolved. The percentages reported in the Results and Discussion sections use different denominators, leading to confusion. Additionally, the total number of isolates listed (n=76) does not match the number of positive animals (n=77), and there's a discrepancy in the total count of T. mentagrophytes strains between different parts of the manuscript (16 vs. 17).

Response 1: Thank you for pointing this out. To represent the percentages of positivity, the denominators have been standardized and always referred to the total number of animals investigated (Abstract, Results, and Discussion sections). The total count of T. mentagrophytes strains is 17 because one animal showed a coinfection with M. canis (see the last column of Figure 5): this has been better explained in lines 265-271. Therefore, the total number of positive animals is 77 and it appears to be correctly written or listed throughout the text or figures. However, a total of 78 isolates were obtained but this number was not explicitly reported in the manuscript to avoid confusion.

Comments 2: A taxonomic error was also found: the common names for the Martes species are incorrectly listed. Martes foina is the beech marten, and Martes martes is the pine marten. These must be corrected to ensure scientific accuracy.

Response 2: Thank you for pointing this out. We apologize for the error: we inadvertently reversed the taxonomic attribution of the species (lines 155-156).  

Comments 3: Finally, the manuscript states that the study was conducted until 2024. This is a future date and needs to be corrected to the actual end date of the study period.

Response 3: Unfortunately, we do not understand this comment, since the study was conducted through 2024 as stated in the title, and was submitted for your review in 2025. However, we noticed that the study involved fifteen years, therefore we changed the study period from “fourteen-year” to “fifteen-year” (lines 3, 250, and 439).   

Detailed comments

Comments 4: The manuscript is comprehensive but could be more concise. Many sentences are long and complex, making the text difficult to follow. I suggest breaking them down into shorter, more direct sentences.

Response 4: Thank you for pointing this out. We tried to improve the text with shorter and more direct sentences, particularly in the Introduction, Discussion, and Conclusion sections.

Comments 5: Some word choices and phrasing also need revision for a more formal, scientific tone, such as replacing "vital players" with "crucial actors." There are also typos to be corrected, including "out-breaks" and "ivestock."

Response 5: Thank you for your suggestions. The phrase containing “vital players” has been changed as well as many others throughout the text to improve it (please, see response 4). The word “livestock”, understood as the collective term for farm animals, appears to be correctly written throughout the text.

Comments 6: In the Materials and Methods section, there's a major point to be corrected: the inversion of the common names for the Martes species, where Martes foina is the beech marten and Martes martes is the pine marten.

Response 6: Please, see Response 2.

Comments 7: The linguistic precision also needs attention, with a revision of phrases like "were collected basing on" and a substitution of the verb "conferred."

Response 7: The suggestions have been accepted (lines 169 and 189). Moreover, we tried to improve the English language throughout the text.  

Comments 8: The term "grattacapo" should be translated or explained.

Response 8: The “grattacapo” brush is a scalp brush used in beauty salons to massage the scalp. In our context, it was used to non-invasively sample animal hair. The term has been translated into the text (line 180).

Comments 9: In the Results, there are significant numerical inconsistencies. The percentages for species are based on the total number of animals (704), while the overall positivity rate is based on non-ruminant animals (593). It is crucial to use a consistent denominator.

Response 9: Thank you for pointing this out. The overall positivity rate previously based on non-ruminant animals (593) has been corrected in the Results section (line 252).  

Comments 10: The number of isolates (76) does not correspond to the number of positive animals (77), and the count of T. mentagrophytes strains (16) conflicts with the number used for further analysis (17). Please clearly differentiate between "positive animals" and the "number of isolates."

Response 10: Please, see Response 1.

Comments 11: In the Discussion and Conclusion, the date inconsistency (study conducted until 2024) must be corrected.

Response 11: Please, see Response 3.

Comments 12: The percentages in the discussion do not match those in the results.

Response 12: Please, see Response 1.

Comments 13: The language, when presenting hypotheses, should be more cautious (for example, "overcrowding may have..." instead of "probably overcrowding").

Response 13: Thank you for pointing this out. The suggestions have been accepted (line 370).  

Comments 14: The conclusion is long and repeats points already covered; please make it more concise and focused on the key takeaways.

Response 14: Thank you for your suggestion: we tried to make the Conclusion section more concise.  

Response to Comments on the Quality of English Language

Point 1: The English is fine and does not require any improvement.

Response 1: Thank you for your opinion.

Additional clarifications

Not applicable

Reviewer 2 Report (New Reviewer)

The Introduction adequately contextualizes the study and discusses the importance of studying wildlife as bioindicators for diseases, particularly zoonoses. They make an effective argument for monitoring dermatophyte distribution and incidence.

The authors appropriately report the number of animals sampled. There appears to have been minor edits to provide more information on the types of fungal-induced lesions seen. The sample collection procedure and the evaluations for terbinafine resistance/susceptibility are sufficiently detailed.

The results are described in sufficient detail and clarity. The additional details on the strains and molecular variations are appreciated.

The discussion adequately explains the findings, discussing details on the identified fungal species and relative distribution of fungi between the sampled animal species.

Overall, the authors do a commendable job of gathering sufficient data across several years and species to provide this information on the distribution of dermatophytes in local wildlife. Such information is essential for monitoring both the health of the animal species and the risk of humans being exposed to zoonotic dermatophytes.

I would be interested to know if there were any differences in the distribution of the fungal species between Umbria and Marche regions, although this may be skewed given that most of the grey squirrel sample population was taken from Umbria.

Brief corrections should be made:

Line 81-83: This sentence (“Unlike animals, humans lack fur, hence an infection by zoophilic and geophilic dermatophytes may induce severe inflammations”) is too similar to that of the paper it is citing (“Unlike animals, humans lack fur, hence an infection by zoophilic or geophilic dermatophytes—outside of their natural habitat—may induce severe inflammations”). Please rephrase this sentence to differentiate it from the reference.

Line 332: Rabbits are classified as lagomorphs, not rodents. The authors should either choose a different animal species as an example or change “Rodents” to a different word or phrase (e.g., “Small wildlife species” or something similar).

Author Response

Thank you very much for taking the time to review this manuscript. We kindly appreciate your opinion. Please find the detailed responses below and the corresponding revisions/corrections highlighted/in track changes in the re-submitted files.

Point-by-point response to Comments and Suggestions for Authors

Detailed comments

Comments 1: I would be interested to know if there were any differences in the distribution of the fungal species between Umbria and Marche regions, although this may be skewed given that most of the grey squirrel sample population was taken from Umbria.

Response 1: Thank you for your interesting comment. We wished to emphasize that we did not comment on the geographic distribution between Umbria and Marche for two reasons: first, as reported in lines 410–414, all grey squirrels were captured in Umbria; second, because the public health institution in which we work is bi-regional and covers the central Italy area. With this contribution, we therefore intend to provide data for a portion of Italian territory (central Italy) rather than from a single region.  

Comments 2: Line 81-83: This sentence (“Unlike animals, humans lack fur, hence an infection by zoophilic and geophilic dermatophytes may induce severe inflammations”) is too similar to that of the paper it is citing (“Unlike animals, humans lack fur, hence an infection by zoophilic or geophilic dermatophytes—outside of their natural habitat—may induce severe inflammations”). Please rephrase this sentence to differentiate it from the reference.

Response 2: Thank you for your suggestion. The sentence has been rephrased in lines 92-94.

Comments 3: Line 332: Rabbits are classified as lagomorphs, not rodents. The authors should either choose a different animal species as an example or change “Rodents” to a different word or phrase (e.g., “Small wildlife species” or something similar).

Response 3: Thank you for pointing this out. We apologize for the error. We chose guinea pig as a different animal species belonging to rodents (line 356).

Response to Comments on the Quality of English Language

Point 1: The English is fine and does not require any improvement.

Response 1: Thank you for your opinion.

Additional clarifications

Not applicable

Reviewer 3 Report (New Reviewer)

The authors conducted surveillance of dermatophytes in wild fauna from the Umbria and Marche regions between 2010 and 2024. A total of 704 animals were examined, and dermatophytes were detected in 77 individuals. The main strengths of the study are the long sampling period and the relatively large number of animals tested. Another strong point is the combination of classical culture-based methods with sequencing for identification, which improves reliability. The focus on uncommon species (e.g., P. mirabile) is also noteworthy, as this provides valuable insights into potential zoonotic risks.

Unfortunately, the manuscript contains several shortcomings and inconsistencies.
First, in the abstract, the authors report frequencies (%) calculated against the total number of animals tested (n = 704), whereas in the results section they use percentages based on positive samples only (n = 77). For example, Paraphyton mirabile is described as 5.96% in the Abstract (42/704) but as 54.54% in the Results (42/77). Such inconsistencies create confusion and hinder interpretation.

Another issue is the use of different identification methods for different sample types. For wild ruminants, direct DNA extraction and hemi-nested PCR were applied. These tests suggested the absence of dermatophyte DNA in ruminants. However, the manuscript does not explain how this method was validated, whether positive controls were included, or whether its sensitivity and specificity are comparable to the standard workflow (culture → PCR). Without such validation, false negatives cannot be ruled out, and conclusions about the absence of dermatophytes in ruminants may be misleading.

The authors also stress the importance of monitoring antifungal resistance in wildlife. However, only genotypic assays were performed, targeting known SQLE mutations. Such tests do not always fully correlate with phenotypic resistance (MIC values) and may fail to detect novel mutations. A major weakness of the study is therefore the lack of susceptibility testing using the broth microdilution method. MIC data would be essential to describe the distribution of resistant isolates and to monitor the emergence of antifungal resistance in wildlife.

Furthermore, the article lacks a supplementary table with detailed information for each isolate: sample ID, collection date, location (region/municipality or coordinates), host species, identification method, ITS result, and where relevant Trichophyton genotype. Without such data, the results are incomplete and difficult to independently verify.

In conclusion, while the dataset is interesting and has potential, the numerous methodological shortcomings, inconsistencies in reporting, and lack of detailed data raise concerns. In its current form, I am not convinced that this manuscript meets the standards of a publishable scientific article.

I have no further detailed comments on the manuscript.

Author Response

Point-by-point response to Comments and Suggestions for Authors

Major comments

Comments 1: First, in the abstract, the authors report frequencies (%) calculated against the total number of animals tested (n = 704), whereas in the results section they use percentages based on positive samples only (n = 77). For example, Paraphyton mirabile is described as 5.96% in the Abstract (42/704) but as 54.54% in the Results (42/77). Such inconsistencies create confusion and hinder interpretation.

Response 1: Thank you for pointing this out. We apologize for the inconvenience. To represent the percentages of positivity, the denominators have been standardized and always referred to the total number of animals investigated (Abstract, Results, and Discussion sections).

Comments 2: Another issue is the use of different identification methods for different sample types. For wild ruminants, direct DNA extraction and hemi-nested PCR were applied. These tests suggested the absence of dermatophyte DNA in ruminants. However, the manuscript does not explain how this method was validated, whether positive controls were included, or whether its sensitivity and specificity are comparable to the standard workflow (culture → PCR). Without such validation, false negatives cannot be ruled out, and conclusions about the absence of dermatophytes in ruminants may be misleading.

Response 2: We would clarify that we used direct DNA extraction and hemi-nested PCR for wild ruminants given the unavailability of Dermasel Agar supplemented with thiamine and inositol, required from typical dermatophytes species isolated in those animals, such as Trichophyton verrucosum (lines 198-203). Regarding the reliability of the PCR assay, it should be emphasized that in every analytical session both positive and negative controls were included during extraction and amplification phases (information added in lines 208-211). Furthermore, given that molecular diagnostic techniques are known to possess high sensitivity and specificity, we consider our negative results to be reliable and unlikely to be false negatives. Indeed, for a subset of ruminants, we initially performed both culture-based assays and PCR in parallel, achieving 100% concordance; however, these data were not included in the manuscript.

Comments 3: The authors also stress the importance of monitoring antifungal resistance in wildlife. However, only genotypic assays were performed, targeting known SQLE mutations. Such tests do not always fully correlate with phenotypic resistance (MIC values) and may fail to detect novel mutations. A major weakness of the study is therefore the lack of susceptibility testing using the broth microdilution method. MIC data would be essential to describe the distribution of resistant isolates and to monitor the emergence of antifungal resistance in wildlife.

Response 3: We agree with this comment; hence we had already listed the lack of susceptibility testing using the broth microdilution method among study limitations (lines 414-415). We stated that it was not performed in this study due to the absence of established clinical breakpoints. Moreover, we have improved the text by presenting the study of resistance with more cautious.

Comments 4: Furthermore, the article lacks a supplementary table with detailed information for each isolate: sample ID, collection date, location (region/municipality or coordinates), host species, identification method, ITS result, and where relevant Trichophyton genotype. Without such data, the results are incomplete and difficult to independently verify.

Response 4: Thank you for pointing this out. We have provided a supplementary table (Table S1).

Comments 5: In conclusion, while the dataset is interesting and has potential, the numerous methodological shortcomings, inconsistencies in reporting, and lack of detailed data raise concerns. In its current form, I am not convinced that this manuscript meets the standards of a publishable scientific article.

Response 5: We are very sorry to hear your opinion. We have tried to respond to your comments by explaining our points of view and have done our best to satisfy your requests by incorporating information into the text to improve it. Nevertheless, we still believe that our study is worthy of being shared with the scientific community because it deals with a little‑known topic in a fairly large number of wild animals, despite the limitations declared and stated in the manuscript itself. We thank you in any case for your valuable suggestions.

Response to Comments on the Quality of English Language

Point 1: The English could be improved to more clearly express the research.

Response 1: We tried to improve the English throughout the text.

Additional clarifications

Not applicable

Round 2

Reviewer 1 Report (New Reviewer)

The manuscript reports a 15-year surveillance study (2010–2024) of dermatophytoses in wildlife from Umbria and Marche, Italy, using a One Health framework. A total of 704 animals were assessed via combined passive and targeted active surveillance. Dermatophytes were detected in 77 cases, with Paraphyton mirabile predominating alongside Trichophyton mentagrophytes complex, Microsporum canis, and several geophilic Arthroderma species. One mixed infection (T. mentagrophytes + M. canis) occurred in a red squirrel. Molecular typing within the T. mentagrophytes complex identified var. interdigitale genotype II and var. mentagrophytes genotype III. All 11 Trichophyton strains tested with a genotypic assay were susceptible to terbinafine. The work underscores wildlife as sentinels for zoonotic dermatophytes, documents non-conventional species in free-ranging hosts, and establishes a regional baseline for antifungal resistance monitoring. Limitations are acknowledged (regional focus, sampling concentration in grey squirrels, absence of MICs), but the design, methods, and findings offer solid, practical contributions to public health and veterinary surveillance.

The manuscript offers a clear, well-structured, and methodologically solid account of wildlife dermatophytoses in Umbria and Marche over a 15 year period, framed within a One Health perspective. The aims are explicit from the outset, the motivation is timely given intensifying human wildlife interfaces, and the ecological and public health rationales are convincingly articulated. The mixed surveillance design, combining passive submissions from rescue centers, veterinary hospitals, and necropsies with targeted active efforts, adds ecological realism and allows the authors to capture both symptomatic and asymptomatic carriers, which is essential for understanding dermatophyte circulation in free-ranging hosts. The diagnostic workflow is appropriate to the objectives: cultural identification complemented by ITS sequencing ensures species-level resolution across most isolates, while the targeted SNP approach within the Trichophyton mentagrophytes complex meaningfully refines taxonomic assignment and links genotypes to host species and clinical presentations. The addition of genotypic screening for terbinafine resistance in Trichophyton provides valuable baseline evidence of susceptibility in the wildlife context and fills a known knowledge gap.

The results are clearly presented and emphasize findings with genuine surveillance value. The predominance of Paraphyton mirabile in grey squirrels is epidemiologically interesting and raises relevant ecological questions about its reservoir status and zoonotic potential. The documentation of T. mentagrophytes complex, distinguishing var. interdigitale genotype II and var. mentagrophytes genotype III, is particularly informative, given the recognized differences in host associations and potential clinical severity. The detection of Microsporum canis in foxes and a coinfected red squirrel aligns with known zoonotic pathways and underscores the need for handler protection and public awareness in peri-urban environments. The narrative sensibly connects species detections to management implications for wildlife rescue centers and veterinarians, reinforcing practical value beyond academic interest.

The discussion is balanced and transparent about constraints inherent to wildlife surveillance. The geographic focus, the uneven temporal and taxonomic distribution driven by program logistics, and the absence of MIC determinations with standardized clinical breakpoints are appropriately acknowledged without undermining the study’s conclusions. Importantly, the manuscript situates the data within broader One Health priorities, linking wildlife sentinel value to early detection of zoonotic risks and to antifungal stewardship concerns. The figures and descriptive statistics guide the reader effectively through host-pathogen patterns over time, and the inclusion of genotype information where available elevates the scientific contribution.

In sum, the manuscript delivers a substantive, well-argued regional baseline on wildlife dermatophytes, integrates molecular epidemiology with practical surveillance, and communicates actionable insights for public and veterinary health. It makes a meaningful contribution to the literature on wildlife mycoses and One Health surveillance and merits publication as presented.

Author Response

Thank you very much for taking the time to review this manuscript. We appreciate your comments about it, despite the limitations you pointed out and those mentioned by the authors in the text.

Reviewer 3 Report (New Reviewer)

The manuscript addresses an interesting topic and presents a large dataset; however, the way it is conducted and analyzed does not allow for a reliable interpretation of the results.
The lack of validation of the molecular method and the absence of phenotypic susceptibility testing disqualify the study in its current form.

Despite the interesting subject and its relevance to public health, the manuscript in its present state could mislead readers regarding the actual prevalence of dermatophytes and the status of terbinafine resistance in wild animal populations.

The authors applied different methodological approaches for different groups of animals — direct DNA extraction and hemi-nested PCR were used for ruminants, while classical culture-based methods were employed for the remaining species. The statement that the obtained negative results are “reliable and unlikely to be false negatives” has no experimental basis. Moreover, the authors mention 100% concordance between PCR and culture results for a certain “subset” of samples, yet these data are not presented in the manuscript. As a result, the conclusions about the absence of dermatophytes in ruminants are unfounded.

The authors claim to have investigated terbinafine resistance, but they used only the DermaGenius PCR assay to detect known mutations in the SQLE gene. No microdilution tests (e.g., according to CLSI) were performed to determine MIC values, which makes it impossible to confirm phenotypic susceptibility. The justification based on the “lack of established clinical breakpoints” is incorrect  in epidemiological studies, raw MIC values can be presented without clinical interpretation.
Consequently, the data on “terbinafine-sensitive strains” are not reliable and cannot serve as a basis for conclusions about the absence of antifungal resistance in wild animals.

The conclusions are too categorical in relation to the data obtained, the statement that “wild fauna does not seem involved in the antifungal resistance phenomenon” is scientifically unjustified.

Author Response

Point-by-point response to Comments and Suggestions for Authors

Detailed comments

Comment 1: The authors applied different methodological approaches for different groups of animals — direct DNA extraction and hemi-nested PCR were used for ruminants, while classical culture-based methods were employed for the remaining species. The statement that the obtained negative results are “reliable and unlikely to be false negatives” has no experimental basis. Moreover, the authors mention 100% concordance between PCR and culture results for a certain “subset” of samples, yet these data are not presented in the manuscript. As a result, the conclusions about the absence of dermatophytes in ruminants are unfounded.

Response 1: The direct application of hemi-nested PCR to samples from wild ruminants is contingent upon the fact that the laboratory does not routinely use a culture medium supplemented with thiamine and inositol, which are necessary for the growth of T. verrucosum, the dermatophyte species most commonly found in these animals. The technical operating procedure employed in the laboratory, following the criteria of the Accredia quality system (the Italian accreditation system), has been previously validated using known positive and known negative samples. To confirm the robustness of the PCR result, each analytical session is intrinsically validated by including positive and negative controls.

Comment 2: The authors claim to have investigated terbinafine resistance, but they used only the DermaGenius PCR assay to detect known mutations in the SQLE gene. No microdilution tests (e.g., according to CLSI) were performed to determine MIC values, which makes it impossible to confirm phenotypic susceptibility. The justification based on the “lack of established clinical breakpoints” is incorrect  in epidemiological studies, raw MIC values can be presented without clinical interpretation. Consequently, the data on “terbinafine-sensitive strains” are not reliable and cannot serve as a basis for conclusions about the absence of antifungal resistance in wild animals.

Response 2: Given that the number of samples tested for susceptibility/resistance to terbinafine is rather small, we accept your comment and have decided to remove from the manuscript the section relating to this aspect. Thank you for prompting us to reflect on the topic and encouraging us to implement this activity in our laboratories. 

Comment 3: The conclusions are too categorical in relation to the data obtained, the statement that “wild fauna does not seem involved in the antifungal resistance phenomenon” is scientifically unjustified.

Response 3: As reported in Response 2, the discussion has been revised by removing the section pertaining to terbinafine resistance.

This manuscript is a resubmission of an earlier submission. The following is a list of the peer review reports and author responses from that submission.

Round 1

Reviewer 1 Report

Line 83: The authors should include some studies that demonstrate the impact of dermatophyte infection in wild animals that affects humans.

The sites where the animals were sampled, specifically for squirrells, are not described.

The sites where the necropsy procedures were performed are not described.

The transportation of the samples to the laboratory and whether any substance was used as a transport medium are not described.

The target used in the study is not described.

The authors must show the years with the highest and lowest number of samples analyzed, as well as their trend. This is an aspect that can be discussed later.

The authors do not present the results of the two PCRs used: sample positivity, identified species, etc.

The conclusions should be rewritten; although important elements are shown, they are written as results.

The limitations of the study must be written.

In summary, some aspects of the text should be improved: the materials and methods section should be better described, the results should be more specific considering the techniques used, the results of the years with the cases presented should be more exploited, and the conclusions should be rewritten.

The article entitled "Wildlife dermatophytoses in Central Italy (Umbria and Marche regions): a fourteen-year investigation (2010–2024)" by Crotti S et al is interesting. However, it needs some important changes before publication.

Major  comments

Line 83: The authors should include some studies that demonstrate the impact of dermatophyte infection in wild animals that affects humans.

The sites where the animals were sampled, specifically for squirrells, are not described.

The sites where the necropsy procedures were performed are not described.

The transportation of the samples to the laboratory and whether any substance was used as a transport medium are not described.

The target used in the study is not described.

The authors must show the years with the highest and lowest number of samples analyzed, as well as their trend. This is an aspect that can be discussed later.

The authors do not present the results of the two PCRs used: sample positivity, identified species, etc.

The conclusions should be rewritten; although important elements are shown, they are written as results.

The limitations of the study must be written.

In summary, some aspects of the text should be improved: the materials and methods section should be better described, the results should be more specific considering the techniques used, the results of the years with the cases presented should be more exploited, and the conclusions should be rewritten.

Minor comments

The abstract must include the number of positive samples.

The keywords dermatophytes and terbinafine resistance must be included.

Line 61: The authors must include a reference.

Line 311: The phrase "In the Authors' opinion" should be deleted.

Line 322 is the objective of the study, not a conclusion. Please rewrite this.

Author Response

Thank you very much for taking the time to review this manuscript. Please find the detailed responses in the attached file and the corresponding revisions/corrections highlighted/in track changes in the re-submitted manuscript.

Reviewer 2 Report

  1. Overall evaluation

Based on the long-term surveillance (2010-2024), this study revealed the diversity of dermatophytes and potential public health risks in wild animals in central Italy, and filled the gap in the surveillance of wild animal mycoses. The study design combined passive and active surveillance, the methodology was comprehensive, and the data were regionally representative. However, there is room for improvement in sample selection, depth of data analysis, and academic rigor of discussion. The following are specific comments and suggestions.

Ii. Main deficiencies and modification suggestions

  1. Study design and methodology

Sample selection bias: grey squirrel accounted for 37.5% (264/704), which may cause the results to overreflect the infection status of this species and underrepresent other species (such as red fox and badger).

Methodological Limitations: Wild ruminants were used for direct DNA extraction (uncultured), but no explanation was given as to why the culture step was skipped, nor was it discussed that this method might miss commensal bacteria or risk of contamination.

Insufficient sample size for drug resistance detection: only 11 T. mentagrophytes strains were tested for drug resistance, and 5 strains were not tested due to insufficient DNA, which may affect the reliability of the conclusion.

  1. Data analysis and result presentation

Missing statistical analysis: No significance tests (such as chi-square test or Fisher's exact test) were performed for infection rates in different species, years, or geographic regions, resulting in a lack of statistical support for the conclusions.

Incomplete information in the figures: margins of error or confidence intervals are not labeled in Figures 1-5, and the sharp drop in 2020 in Figure 4 (annual distribution) is unexplained (possibly related to COVID-19 but not mentioned).

Insufficient analysis of co-infection cases: there was only one case of co-infection of T. mentagrophytes and M. canis in red squirrels, but the epidemiological significance or potential synergistic mechanism was not explored.

  1. Discussion and literature support

One-sided literature citation: insufficient comparison of similar studies. "For example, the first detection of P. mirabile in hedgehog has not been compared with data from other European regions, so it is difficult to determine whether it is a regionally specific phenomenon."

Insufficient discussion of public health implications: although frequent contact between grey squirrels and humans is mentioned, no quantification of risks (e.g. association of infection rates with human cases) or specific prevention and control recommendations are made.

Resistance discussion Brief: Implications of this study were not analyzed in the context of global resistance trends, such as the rising rate of T. mentagrophytes resistance in Asia.

  1. Writing and formatting

Inconsistent terminology: Some species names are not italicized (e.g., "Arthroderma curreyi"), so a uniform format is required.

Chart readability: The species names in Figure 2 do not correspond intuitively to the bar chart (e.g. "Eurasian beaver 1" requires location).

Iii. Other suggestions

In the discussion section, the authors should address the following points:

Additional study limitations should be specified: The limitations of the geographical scope of the samples (restricted to central Italy) and species coverage should be clearly stated to avoid overgeneralization of the conclusions.

Expanded future research directions: It is recommended to conduct molecular epidemiological tracing (e.g., strain typing) or studies on host-fungal interactions.

Conclusion

This study provides valuable data for monitoring dermatophytosis in wild animals, but further improvements are needed in methodological transparency, depth of data analysis, and rigor of academic discussion. By incorporating additional statistical analyses, refining figures and tables, expanding the comparative literature review, and clarifying study limitations, the paper’s academic value and impact will be significantly enhanced.

For the fungal population on animal surface, it is recommended to use metagenomic sequencing combined with ITS sequencing after isolation and culture.

Author Response

(The authors gave the same response as above.)

Reviewer 3 Report

This manuscript presents relevant and consistent data on the occurrence of dermatophytes in wild animals from the Umbria and Marche regions (Italy) over 14 years. The sampling was extensive, the laboratory methods were sound, and the study aligned well with the One Health framework. The combination of passive and active surveillance and the use of PCR for fungal identification is a clear strength of this study.

However, a few critical points need to be addressed or at least acknowledged to clarify the actual scope and limits of the findings. The authors suggest a risk related to dermatophytes in anthropized environments; however, no spatial data are presented to support this link. There was no map, georeferencing, or basic classification of where the animals were collected (urban, peri-urban, rural). I fully understand that working with wildlife comes with logistical challenges, origins are often uncertain, and the data may be fragmentary. However, if any spatial information is available, its inclusion would considerably strengthen the manuscript's ecological framing and One Health implications.

The second point concerns the partial nature of the antifungal resistance assessment. Only T. mentagrophytes strains were tested for terbinafine susceptibility using the DermaGenius® real-time PCR assay, which detects known mutations in the SQLE gene. While this is a valid genotypic screening tool, it has limitations: it does not detect resistance mechanisms outside the target mutation set or evaluate actual phenotypic susceptibility. Importantly, the most prevalent fungal species in this study, Paraphyton mirabile, Microsporum canis, and several Arthroderma spp., were not tested. Even attempting to apply the kit and reporting a negative or failed detection would have added value, possibly indicating genetic divergence or assay limitations. The manuscript should have acknowledged this absence as a technical limitation or a scope constraint.

Additionally, because all isolates were cultured, it would theoretically have been possible to apply broth microdilution MIC testing (CLSI M38-A) to assess terbinafine susceptibility in all species. I fully recognize the practical difficulties, especially with slow-growing or unstandardized organisms and the lack of clinical breakpoints. However, it remains a route that could be discussed or mentioned as an alternative approach.

Although the authors do not claim to present an epidemiological study, the data collected (sample size and species-specific infection rates) could support a more structured interpretation. Statements regarding zoonotic risk and environmental surveillance would benefit from being grounded in a more clearly defined spatial and analytical context.

The manuscript refers to a One Health perspective but has not been developed beyond wildlife surveillance. If the authors keep this framing, it would be appropriate to clarify how the results contribute to One Health or adjust the wording to reflect the study’s actual scope.

In summary, this is a well-executed study with meaningful contributions. The manuscript is well written; however, the authors should openly acknowledge the study's limitations, particularly regarding resistance in the most prevalent species and the ecological origin of the animals. Clarifying these boundaries will strengthen the manuscript.

The first sentence of the abstract should be rephrased. This is somewhat generic and could better reflect the study's primary objective.

The manuscript returns more than once to the theme of wildlife as sentinels and as potential zoonotic reservoirs. A more concise presentation of these points could improve the overall flow and sharpen the central arguments.

A table summarizing fungal species × host species × lesion presence × resistance tested (yes/no) would enhance clarity for the readers.

In the methods section, please specify that the resistance test was not phenotypic and clarify its SQLE mutation-based scope.

The authors should consider briefly stating in the Discussion that CLSI M38A could be an alternative method for future resistance analysis, even if it is not feasible for this study.

Key concepts such as One Health, zoonotic risk, and asymptomatic carriers appeared repeatedly across sections. I recommend streamlining to maintain the focus. It would also be appropriate to briefly mention the ecological uncertainty regarding the classification of P. mirabile as either zoophilic or geophilic.

The current conclusion reads more like an extension of the discussion or introduction than a synthesis of the study’s key findings. I suggest reformulating it to concisely highlight the main results, acknowledge the study’s limitations (e.g., resistance testing restricted to T. mentagrophytes and lack of spatial data), and clearly state the implications for surveillance and future research.

Author Response

(The authors gave the same response as above.)

Round 2

Reviewer 1 Report

The authors do not mention the PCR target (lines 179-182 ). They also do not describe why they use nested PCR.
It is well known that nested PCR has high risks of contamination. The authors should explain what measures they used to prevent this?.

Line 269 What were the causes of DNA being unavailable?
The authors do not explain the annual distribution of the animals. This aspect is vitally important because it can set trends for other studies.
The discussion is limited to discussing their results, but does not compare them with those described in the literature.
The discussion can be expanded and updated with the suggested bibliography.

The authors do not mention the PCR target (lines 179-182 ). They also do not describe why they use nested PCR.
It is well known that nested PCR has high risks of contamination. The authors should explain what measures they used to prevent this?.

Line 269 What were the causes of DNA being unavailable?
The authors do not explain the annual distribution of the animals. This aspect is vitally important because it can set trends for other studies.
The discussion is limited to discussing their results, but does not compare them with those described in the literature.
The discussion can be expanded and updated with the suggested bibliography.

Author Response

Comments 1: The authors do not mention the PCR target (lines 179-182 ). They also do not describe why they use nested PCR.

Response 1: Thank you for pointing this out. In lines 181-183 the PCR targets were mentioned.

Comments 2: It is well known that nested PCR has high risks of contamination. The authors should explain what measures they used to prevent this?

Response 2: Thank you for pointing this out. A negative amplification control has been included in each analytical session with the aim to evaluate the absence of contamination. Moreover, the specificity of the nested PCR was ensured by the Sanger sequencing which identified dermatophytes species for each positive sample. 

Comments 3: Line 269 What were the causes of DNA being unavailable?

Response 3: The DNA was no longer available because it was tested in other researches and then accidentally discarded.

Comments 4: The authors do not explain the annual distribution of the animals. This aspect is vitally important because it can set trends for other studies.

Response 4: Unfortunately, the annual distribution of the animals was not regular due to the lack of regular active surveillance plans that had repercussions both on the wild species enrolled and on the number of samples analyzed per year. In the Authors opinion this concept was anyway explained in the manuscript (e.g. “Discussion” section).

Comments 5: The discussion is limited to discussing their results, but does not compare them with those described in the literature.

Response 5: Thank you for pointing this out. We improved the “Discussion” section by adding comparison with literature information in lines 315-317 and 332-335.

Comments 6: The discussion can be expanded and updated with the suggested bibliography.

Response 6: Thank you for your suggestion. The discussion has been improved with references 29 and 31.

Round 3

Reviewer 1 Report

I only have minor comments Linea 67: Substitute is relatively rare but is not as frequently described. Please see you Gupta AK, Wang T, Susmita, Talukder M, Bakotic WL. Global Dermatophyte Infections Linked to Human and Animal Health: A Scoping Review. Microorganisms. 2025 Mar 3;13(3):575. doi: 10.3390/microorganisms13030575. Reference 10 must be replaced by Gupta AK, Wang T, Susmita, Talukder M, Bakotic WL. Global Dermatophyte Infections Linked to Human and Animal Health: A Scoping Review. Microorganisms. 2025 Mar 3;13(3):575. doi: 10.3390/microorganisms13030575.

Line 71 - 79: A reference must be included in that paragraph.

Line 152: Needs to include a reference.

Line 198: Since the authors use a nested PCR and this technique carries a risk of contamination, I recommend that the materials and methods section, when explaining the technique, include a sentence stating that all measures were taken to prevent contamination.

Line 359-360: The expression: as shown in figure 2 must be deleted.

Line 360-363 is a conclusion not derived from this study. The authors should rewrite that conclusion according to their results. References 17 and 18 should be reviewed. There may be problems in the text.  

Author Response

Comments 1: Linea 67: Substitute is relatively rare but is not as frequently described. Please see you Gupta AK, Wang T, Susmita, Talukder M, Bakotic WL. Global Dermatophyte Infections Linked to Human and Animal Health: A Scoping Review. Microorganisms. 2025 Mar 3;13(3):575. doi: 10.3390/microorganisms13030575. Reference 10 must be replaced by Gupta AK, Wang T, Susmita, Talukder M, Bakotic WL. Global Dermatophyte Infections Linked to Human and Animal Health: A Scoping Review. Microorganisms. 2025 Mar 3;13(3):575. doi: 10.3390/microorganisms13030575.

Response 1: Thank you, your suggestions have been received and reference 10 has been substituted.

Comments 2: Line 71 - 79: A reference must be included in that paragraph.

Response 2: References 11 and 12 have been added in lines 73 and 79, respectively.

Comments 3: Line 152: Needs to include a reference.

Response 3: References 16 has been added in line 152.

Comments 4: Line 198: Since the authors use a nested PCR and this technique carries a risk of contamination, I recommend that the materials and methods section, when explaining the technique, include a sentence stating that all measures were taken to prevent contamination.

Response 4: A sentence has been added in lines 199-200.

Comments 5: Line 359-360: The expression: as shown in figure 2 must be deleted.

Response 5: The expression “as shown in figure 2” has been deleted.

Comments 6: Line 360-363 is a conclusion not derived from this study. The authors should rewrite that conclusion according to their results.

Response 6: Thank you for pointing this out. The sentence has been rewritten.

Comments 7: References 17 and 18 should be reviewed. There may be problems in the text.  

Response 7: reference 17 and 18 become 20 and 21, respectively. The link available in reference 17 allows you to open the correct window. Reference 21 has been reviewed.
